# Sparsity Reveals Strangers: A Sparse Auto Encoder Approach to OOD Detection

## Abstract

Sparse Autoencoder (SAE) has recently proven effective for interpretability in large language models by transforming dense hidden states into sparse, semantically meaningful components. In this work, we extend this paradigm to Vision Transformer (ViT), focusing on the [CLS] token, a compact representation that aggregates global image information but is difficult to analyze directly. By training an SAE on [CLS] tokens, we unfold this compressed signal into a sparse latent space that reveals consistent, class-specific activation patterns for in-distribution (ID) data and distinctive deviations for out-of-distribution (OOD) data. To make this structure explicit, we introduce the Class Activation Profiles (CAPs), which rank SAE latent dimensions by their mean activation for each class, providing a class-conditioned reference that can be used to test in OOD detection. These observations highlight that ID samples not only concentrate activation in a small set of dominant features but also preserve a stable rank hierarchy, whereas OOD samples disrupt this structure. Leveraging this insight, we demonstrate that a simple Spearman rank correlation measure can effectively capture the OOD data. This approach yields competitive AUROC scores and achieves a state-of-the-art FPR95 on one dataset while remaining highly competitive on the others. Notably, performance is steady across different OOD benchmarks, indicating robustness. These findings illustrate that the structural invariants revealed by SAE can be transformed into lightweight analytical tools, highlighting their value not just for detection but also for enhancing the transparency and interpretability of ViT feature representations.

## 1 Introduction

Deep learning has achieved impressive success across a wide range of vision tasks, yet its reliability in open-world scenarios remains a central challenge. Models trained on a fixed distribution often encounter data that lie outside this distribution, known as Out-of-Distribution (OOD) samples. The ability to reliably detect such inputs is a cornerstone of safe and trustworthy AI, preventing unpredictable and potentially harmful outcomes in critical applications like autonomous driving or medical diagnosis (Hendrycks & Gimpel (2016); Hendrycks et al. (2021); Yang et al. (2024)). Robust OOD detection is therefore indispensable for building trustworthy AI systems.

Over the past years, a variety of methods have been proposed for OOD detection. One of the simplest and most widely adopted techniques is the maximum softmax probability (MSP), which measures the confidence of a classifier's prediction and treats low-confidence predictions as potential OOD samples (Hendrycks & Gimpel (2016)). While computationally efficient, however, MSP is prone to overconfidence even for inputs that bear little resemblance to the training data. This has led to the development of more sophisticated techniques that analyze the model's internal feature representations. Yet, with the rise of increasingly complex architectures like Vision Transformer (ViT) (Dosovitskiy et al. (2020)), interpreting these high-dimensional feature spaces has become a significant challenge in itself.

ViTs represent images as sequences of patch tokens and aggregate global information from patch tokens across multiple layers of self-attention through a designated [CLS] token for classification. The [CLS] embedding naturally serves as a compact summary for classification. While this token provides a rich, holistic representation of the input, its dense and entangled nature makes it a "black

box", obscuring the underlying factors that contribute to a decision (Chefer et al. (2021); Xie et al. (2022)). This interpretability gap limits our ability to build robust OOD detection mechanisms directly upon these features.

In this work, we take a significant step towards bridging this gap by drawing inspiration from recent breakthroughs in the interpretability of large language models (LLMs). Researchers have successfully used SAE to decompose complex internal activations of LLMs by expanding the latent dimensionality while enforcing sparsity constraints (Cunningham et al. (2023); Olshausen & Field (1997); Ng et al. (2011)). We posit that a similar approach can be effective for dissecting the [CLS] token in ViTs. Our central hypothesis is that we can obtain more structured and semantically meaningful activation patterns by "unfolding" the dense [CLS] vector into a sparsely activated latent space. This sparse decomposition uncovers a clear, human-interpretable structure in the latent space. For each ID class, only a small, specific set of latent dimensions is characteristically activated. By ordering these dimensions by their activation magnitude, we define **Class Activation Profiles (CAPs)**, class-conditioned templates having a compact "head" of strongly firing features and a long, mostly silent "tail" distribution. During inference, ID samples reliably reproduce this distribution; however, OOD samples do not. These insights motivate testing rank consistency—leading to the use of rank-based measures for OOD detection.

Capitalizing on this discovery, we introduce a simple, computationally efficient OOD scoring function. Without retraining the backbone, it attains competitive AUROC scores and achieves state-of-the-art (SOTA) FPR95 on one standard benchmarks dataset, while remaining robust across a wide range of datasets. An overview of the pipeline appears in Fig. 1. Our contributions are threefold:

**1. Sparse autoencoders for vision interpretability and OOD.** We are the first to successfully adapt and apply SAE to ViT. Trained on [CLS] tokens, our SAE projects dense ViT features into a sparse latent space providing a clear basis for analysis in vision and enabling class-conditioned diagnostics that were previously opaque.

**2. Class Activation Profiles (CAPs) as a structural invariant.** For each class, we rank latent dimensions by ID mean activation to obtain a fixed reference profile used at inference to assess structural conformity.

**3. CAP-derived, interpretable OOD methods with strong and stable performance.** From CAPs, we derive simple rank-based scoring that requires no backbone retraining, is computationally efficient, matches SOTA AUROC, attains a SOTA FPR95 on one dataset while remaining highly competitive on the others.

## 2 RELATED WORKS

Our work is positioned at the intersection of three key research areas: out-of-distribution detection, the interpretability of vision models, and the emerging field of sparse representation learning for large neural networks.

**Out-of-Distribution (OOD) Detection.** The problem of detecting when a model encounters inputs outside its training distribution has been extensively studied in recent years. A widely recognized baseline is the MSP, which relies on predictive confidence to identify anomalous samples (Hendrycks & Gimpel (2016)). Building upon this, ODIN introduced temperature scaling and small input perturbations to sharpen confidence separation between ID and OOD data (Liang et al. (2017)). Beyond classifier confidence, distance-based approaches such as Mahalanobis scoring model the distribution of intermediate features and measure distances from class-conditional centroids, yielding more robust separation in many settings. (Lee et al. (2018)). Likelihood-based methods, using generative models, attempt to detect anomalies by assigning probabilities directly to inputs. (Nalisnick et al. (2018)). However, such methods often suffer from the likelihood paradox, where OOD samples may paradoxically be assigned higher likelihoods than true ID samples. Further directions have introduced energy-based scores derived from logits (Liu et al. (2020)), density-ratio estimation (Ren et al. (2019)), and outlier exposure (Hendrycks et al. (2018)), each expanding the methodological toolkit. Despite their effectiveness, these approaches still face limitations in scalability, generalization across diverse OOD conditions, and interpretability. Most operate as post-hoc scoring functions, leaving open questions about the underlying representations that drive OOD sensitivity. While powerful, these methods often treat the feature space as a monolithic entity. Our approach differs by

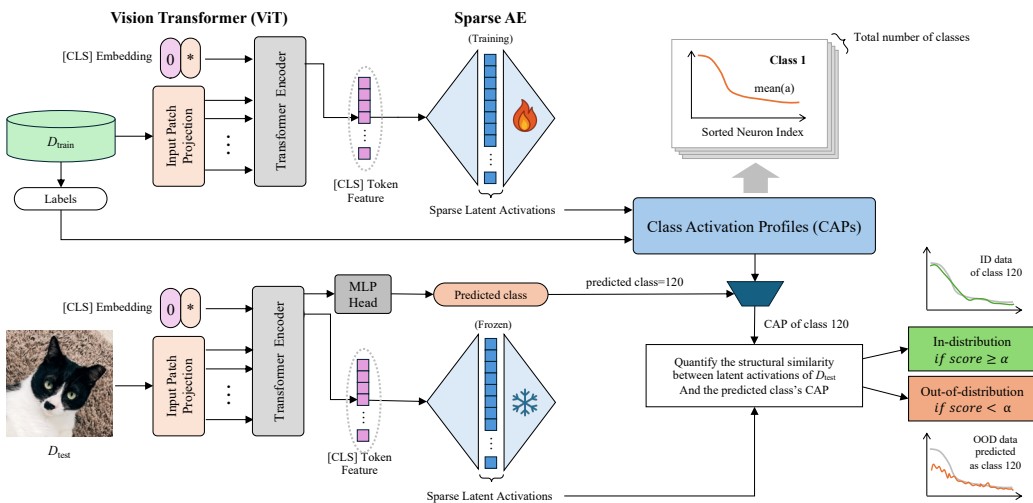

Figure 1: **Overview of our OOD detection method.** In the **(a) Setup Phase**, a SAE is trained on ID data to learn the underlying data structure of the ID data, which is stored as CAPs. In the **(b) Inference Phase**, the latent activation of a test sample is compared to its predicted class's CAP using the final OOD score.

first decomposing this space into a more granular, interpretable basis before defining a detection criterion, allowing us to capture more subtle structural discrepancies.

**Vision Transformers and the [CLS] Token.** The Vision Transformer (ViT) architecture shifts inductive bias away from convolutional locality toward token-based image representations, modeling long-range dependencies through self-attention Dosovitskiy et al. (2020). Within this framework, the [CLS] token plays a central role: it aggregates information from all patch tokens and is directly used for classification. Because of this pivotal function, the [CLS] embedding has become a natural focus for both performance optimization and interpretability studies. Research has shown that the way global information is aggregated through [CLS] competes with alternative pooling mechanisms and directly impacts overall accuracy Touvron et al. (2021). Further analysis of internal ViT features indicates that the [CLS] token encodes not only class-specific semantics but also transferable distribution-level information, suggesting that it serves as more than a simple categorical summary Raghu et al. (2021). At the same time, studies have raised concerns that this representation may act as a bottleneck, with limited expressiveness unless its internal composition is better understood Štefánik et al. (2022); Kan & Shiraishi (2023). Despite these insights, interpreting the [CLS] token remains difficult. Contributions from all patches are densely entangled through multiple layers of self-attention, making it unclear which latent dimensions capture which semantic factors. Visualization tools such as attention rollout and gradient-based attribution provide partial clues about token interactions but fail to expose what specific coordinates of the [CLS] embedding represent Abnar & Zuidema (2020); Chefer et al. (2021). In parallel, most OOD detection methods based on transformers bypass this interpretability challenge by applying generic scoring rules—such as energy-based methods Liu et al. (2020) or virtual logit matching (ViM) Wang et al. (2022)—directly to [CLS] vectors, without uncovering their semantic or structural organization. This gap leaves a pressing question: if the [CLS] token is the decision-critical representation in ViTs, how can its internal structure be reorganized into a more interpretable form that also improves reliability for tasks such as OOD detection? Our work addresses this challenge by applying sparse autoencoders to restructure the [CLS] token into a class-conditioned, sparse feature space, making its activation patterns transparent for in-distribution data and highlighting systematic deviations in OOD inputs.

**Sparse Autoencoder.** Sparse Autoencoder (SAE) was originally proposed as biologically inspired models of efficient coding (Olshausen & Field (1997)), but their applications have expanded significantly in modern machine learning. SAE has become important in natural language processing and large language models (LLMs). For example, recent work demonstrated that sparse autocoding can disentangle directions in embedding spaces of LLMs, producing interpretable units that

correspond to semantic concepts or stylistic attributes (Elhage et al. (2022); Bricken et al. (2023); Cunningham et al. (2023)). This line of research highlights the ability of SAE to expose hidden factors of variation that are otherwise superimposed in dense representations. By expanding the latent dimensionality, SAE provides room to separate features, and by enforcing sparsity, it ensures that only a small subset of dimensions activate meaningfully for a given input. This makes the learned representation easier to interpret, as each dimension is more likely to capture a distinct semantic attribute (Makhzani & Frey (2013)). In practice, SAE has been used to learn disentangled representations in images, speech, and text (Lee et al. (2007)). In the context of interpretability, their most recent impact has been in analyzing foundation models: SAE applied to LLM activations has revealed modular structures in otherwise dense hidden states. This growing evidence suggests that SAE is not only useful for compression or reconstruction but also for making hidden representations human-interpretable. These qualities make them an attractive candidate for analyzing ViT [CLS] tokens, where dense entanglement has long been a barrier to transparency.

Our work represents a novel application of this paradigm: instead of just identifying learned features, we analyze the structural patterns of their activations and leverage these patterns for a downstream task, OOD detection. We are the first to propose and demonstrate that this powerful technique can be successfully transferred from the NLP domain to enhance the robustness of vision models.

# 3 PATTERN ANALYSIS

To devise an effective OOD detection method, we first analyze how the SAE reshapes [CLS] token representations into a structured sparse space. This analysis is not merely preliminary; it grounds the design of our scoring methods in Sec. 4. By training SAE on [CLS] tokens from a pre-trained ViT, dense and entangled features are "unfolded" into a high-dimensional dictionary of sparse, partly disentangled factors. This allows us to (i) characterize how ID concepts are encoded and (ii) observe how OOD samples fail to conform to the learned semantic patterns.

## 3.1 CLASS-SPECIFIC SPARSE SIGNATURES IN ID DATA

We refer to ImageNet (Deng et al. (2009)) that was used to pre-train ViT as ID. Training the SAE on ViT [CLS] tokens pre-trained on ImageNet yields a sparse latent space in which ID classes exhibit consistent and highly discriminative patterns. For each ImageNet class, we observe that a subset of neurons is repeatedly activated in samples from the same class. For each class, core indices are obtained by ranking latent dimensions by their average activation on ID samples and selecting the top 10%. Intuitively, these core indices are those latent dimensions that, on average, show the highest activation for that class across its ID exemplars; they act as canonical detectors for the class's semantic content.

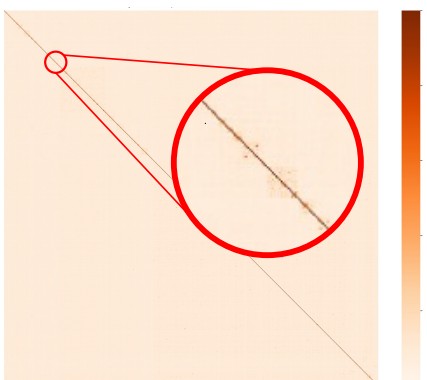

Figure 2: Pairwise Jaccard similarity of top-10% activation indices across 1,000 ImageNet classes. Each cell shows the overlap between the core feature sets of two classes.

To assess how unique these core sets are, we compute the Jaccard similarity coefficient between every pair of 1,000 ImageNet classes, measuring the fraction of overlapping indices relative to the union of their core sets. The resulting heatmap is shown in Fig 2. In this visualization, the red indicates a high Jaccard coefficient, meaning two classes share many core indices, while near white colors correspond to little or no overlap. The heatmap shows a strong diagonal dominance with near-zero off-diagonal values, confirming that most classes have largely unique core indices. Occasional weakly positive off-diagonal entries appear between semantically related categories—for example, dog breeds or cat species—indicating that similar classes can share a small subset of latent dimensions. Overall, however, the clear diagonal pattern demonstrates that the SAE disentangles class representations effectively, yielding a stable and class-conditional baseline for anomaly measurement.

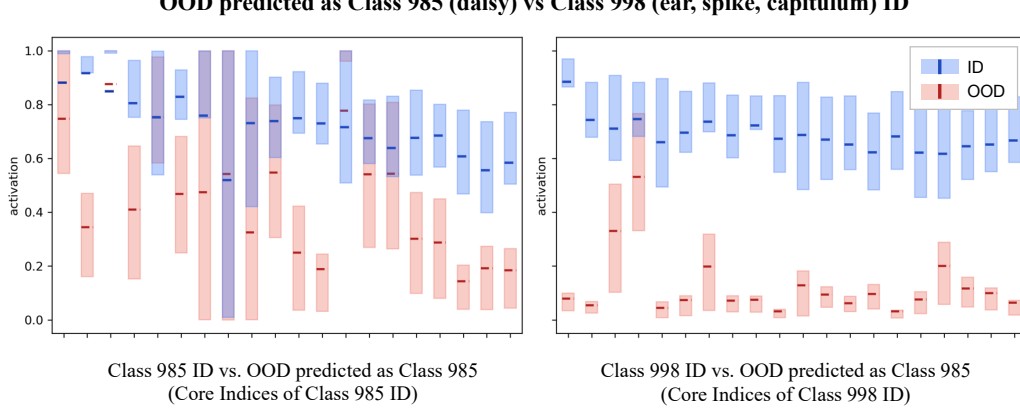

Class 985 ID vs. OOD predicted as Class 985
(Core Indices of Class 985 ID)

Class 998 ID vs. OOD predicted as Class 985
(Core Indices of Class 998 ID)

Figure 3: Activation comparison of ID (blue) and OOD (red) samples on class core indices. The left panel shows OOD samples from iNaturalist that are misclassified into an ImageNet class: these OOD inputs still activate the core indices of their predicted class at relatively high levels. The right panel evaluates the same OOD samples on the core indices of an unrelated class, where activations drop markedly. This illustrates that even misclassified OOD samples tend to follow the core-index pattern of their assigned class, but the strength of activation remains weaker than true ID samples.

## 3.2 ACTIVATION PROFILES OF OOD SAMPLES

Although OOD data were never seen during ViT pre-training, we observe that OOD samples misclassified into an ImageNet class often exhibit specific activation patterns on the core indices of that class. In other words, even when an OOD input is incorrectly classified, it still tends to activate the same subset of latent dimensions that define the predicted ID class. Fig 3. from the iNaturalist (Van Horn et al. (2018)) illustrates this phenomenon. The left panel shows OOD samples (red) predicted as daisy exhibit high activations on the daisy class's core indices. The right panel indicates the same OOD samples evaluated on the core indices of unrelated classes, where activations drop substantially. While OOD inputs do activate the core indices of their predicted class more strongly than those of unrelated classes, their responses remain consistently weaker than those of ID samples. The OOD mean activations at core indices fall below those of their ID counterparts; while OOD samples do activate the core indices, their responses remain consistently weaker than those of ID data. See Appendix A for more examples.

Figure 4 further summarizes this comparison across three groups: ID samples on their own class's core indices (left), OOD samples on the core indices of their predicted class (middle), and OOD samples on the core indices of other classes (right). Results are shown for iNaturalist as the OOD dataset. ID samples exhibit the highest activation on their class-specific core indices. OOD samples, when assigned to a particular class by the ViT, activate the core indices of

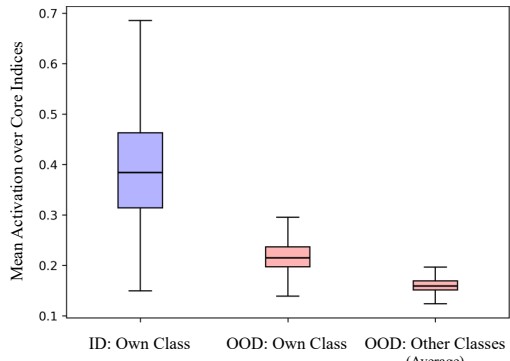

Figure 4: Comparison of mean activations on class-specific core indices for ID and OOD samples from iNaturalist. Left: ID samples show the strongest activation on the core indices of their ground-truth class. Middle: OOD samples, when misclassified into a given class, also activate the core indices of that predicted class but with lower intensity. Right: the same OOD samples show much weaker activations on the core indices of other ID classes.

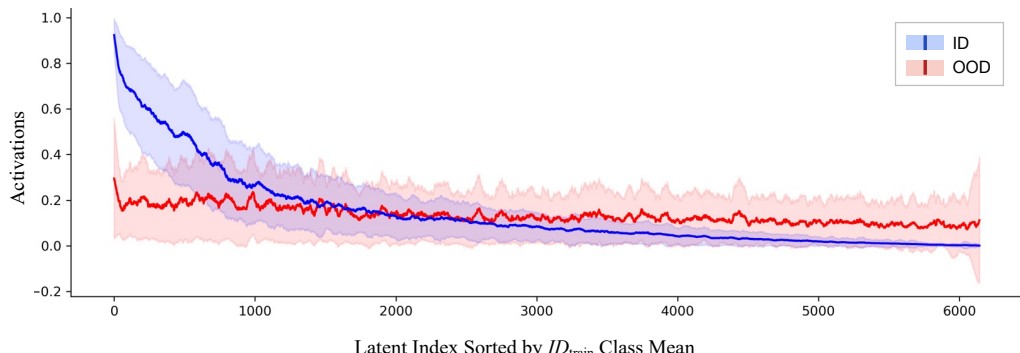

Figure 5: Example activation distribution of an ImageNet class, comparing ID samples (blue) and OOD samples from iNaturalist (red) along the class's CAP-ordered latent dimensions. The x-axis follows the index order defined by the trained ID mean activation (head → tail). ID samples show a sharp, high-energy head followed by a silent tail, while OOD samples exhibit weaker head activations and elevated, noisy tails.

that predicted class but with noticeably lower. The same trend holds across other OOD datasets, as reported in Appendix B.

In summary, while OOD samples do focus on the correct class's core indices, the strength of their activation is systematically weaker and more variable. This attenuation provides a robust, class-conditional signal for distinguishing OOD inputs.

### 3.3 ID vs OOD Activation Pattern Differences

Beyond individual core indices, we analyze the entire activation profile of each class by sorting latent dimensions according to the mean activation of ID training samples. Fig. 5 shows an example from the iNaturalist dataset. Here, ID samples (blue) and OOD samples (red) are plotted along the CAP-ordered dimensions of a single ImageNet class. Two consistent effects appear:

- Higher Head: ID samples produce stronger activations in the top-ranked head features, confirming that they reliably engage the most important latent factors.

- Lower Tail: ID samples exhibit lower activations in the tail region, while OOD samples inflate this region, producing a noisy and diffuse tail.

The differences between ID and OOD are not limited to a few core indices; they appear across the entire sorted profile from the head to the tail. Importantly, the precise shape of these deviations varies across classes, underscoring the need for a class-conditioned perspective. Additional per-class plots for other OOD datasets are provided in Appendix C.

To formalize these observations, we define **Class Activation Profiles (CAPs)** by sorting the latent dimensions of each class according to the mean activation computed from its ID training samples. Each class's CAP captures this standard ID structure and serves as a fixed reference against which test samples can be evaluated. In ID data, this rank hierarchy is stable: a few dominant head features consistently separate themselves with high activations, while tail indices remain firmly suppressed. In OOD data, however, the head becomes flattened and diffuse. Instead of a sharp separation, many head features receive similarly weak activations compared to ID, blurring the fine-grained order within the head. As a result, even when OOD samples activate roughly the same indices as ID, their relative ranks deviate from the class's canonical pattern. This breakdown of ordering naturally motivates a rank-based comparison to quantify how closely a sample preserves the expected hierarchy.

## 4 EXPERIMENTS

### 4.1 EXPERIMENTAL SETUP

**OpenOOD v1.5.** In our experiments, we adopt OpenOOD v1.5, the latest version of the comprehensive OOD detection benchmark framework (Zhang et al. (2023)), to ensure a fair and methodologically consistent evaluation against existing works. We compare the performance of our method with 20 SOTA OOD detection methods. The ID data for all experiments was the ImageNet-1K dataset. The OOD evaluation was conducted using a suite of five datasets, organized into two groups based on their semantic distance to ImageNet-1K: near-OOD (SSB-hard (Vaze et al. (2022)), NINCO (Bitterwolf et al. (2023))) and far-OOD (iNaturalist (Van Horn et al. (2018), Textures (Cimpoi et al. (2014)), OpenImage-O Kuznetsova et al. (2020))). Further details on these benchmark datasets can be found in Appendix E.

**Backbone Architectures.** Our primary experiments are conducted using a pre-trained ViT-B/16 as the backbone, given our method's focus on analyzing the global features captured by the [CLS] token. To verify the generalizability of our approach across different Transformer-based architectures, we further evaluate it on the Swin Transformer (Liu et al. (2021)) and DINOv2 (Oquab et al. (2023). For the evaluation, both ViT-B/16 and Swin Transformer utilized OpenOOD's standard implementations, whereas DINOv2 was newly integrated into the framework by us.

**Sparse Autoencoder.** To construct our CAPs, we train a SAE using feature vectors extracted from the pre-trained backbones: the [CLS] token representation for ViT-B/16 and the output of the final average pooling layer for ResNet-50. The learning objective of the SAE is to accurately reconstruct the input features while simultaneously encouraging sparsity in its latent space activations. To achieve this, the loss function is composed of two terms: the Mean Squared Error (MSE), which measures the Reconstruction Error, and a KL Divergence penalty that guides the latent activations toward a target sparsity level.

$$\mathcal{L}_{\text{SAE}} = \text{MSE}(x, \hat{x}) + \lambda \sum_{j=1}^{d} \text{KL}(\rho \parallel \hat{\rho}_j) \tag{1}$$

$x$ is the original feature vector and $\hat{x}$ is the reconstructed vector. The second term is a regularization term that penalizes the deviation of the actual average activation of a latent neuron($\hat{\rho}_j$) from the target sparsity($\rho$). To determine the optimal hyperparameters for the SAE, we performed a grid search over three key hyperparameters: latent space dimension ($d$), sparsity weight($\lambda$), and the target sparsity($\rho$).

**OOD Scoring.** For the OOD score computation, we select Spearman's Rank Correlation Coefficient since it is robust to outliers and aligns with our core hypothesis that OOD samples exhibit a different activation pattern compared to ID data. Instead of using raw activation magnitudes, this score measures the structural similarity between the activation rank vector of a test sample, $R_a$, and the pre-computed CAPs for its predicted class, $R_c$. A score closer to 1 indicates a high degree of similarity to the normal activation pattern for that class. Further details on the calculation are provided in Appendix F.

**Metrics.** We report using two measures: FPR95, the false positive rate on OOD inputs when the ID true positive rate is fixed at 95% (lower is better), and AUROC, the area under the ROC curve (higher is better).

### 4.2 RESULTS ON VIT

In the upper part of Table 1, we compare the OOD detection performance of our method with several prominent SOTA approaches on a ViT-B/16 backbone using ImageNet-1K as the ID dataset. The AUROC and FPR95 results show that our CAP-based approach delivers highly competitive performance compared to existing benchmarks.

Notably, our method achieves a SOTA FPR95 performance on OpenImage-O, reaching a 29.49% on the ViT backbone. In contrast, methods such as GradNorm Huang et al. (2021) and ASH Djurisic et al. (2022) can fail catastrophically, with FPR95 exceeding 90% on multiple datasets. Our approach is particularly robust on challenging near-OOD data: for instance, on NINCO, we attains 49.27% FPR95, outperforming recent strong baselines including ViM (57.45%) Wang et al. (2022) and SHE (56.01%) Zhang et al. (2022).

| Backbone | Method | SSB-Hard | | NINCO | | iNaturalist | | Textures | | OpenImage-O | |
|---|---|---|---|---|---|---|---|---|---|---|---|
| | | AUROC | FPR95 | AUROC | FPR95 | AUROC | FPR95 | AUROC | FPR95 | AUROC | FPR95 |
| ViT-B/16 | ASH | 53.89 | 93.50 | 52.52 | 95.40 | 50.63 | 97.02 | 48.53 | 98.49 | 55.52 | 94.80 |
| | GEN | 70.09 | **82.24** | 82.51 | 59.31 | 93.54 | 22.94 | 90.23 | 38.31 | 90.27 | 35.43 |
| | GradNorm | 42.96 | 93.62 | 64.40 | 95.81 | 42.42 | 91.16 | 44.99 | 92.25 | 37.82 | 94.52 |
| | KNN | 65.97 | 86.22 | 82.25 | 54.73 | 91.46 | 27.74 | 91.12 | 33.23 | 89.86 | 34.82 |
| | MDS | 71.57 | 83.47 | 86.52 | 48.76 | 96.01 | 20.66 | 89.41 | 38.90 | **92.38** | 30.35 |
| | MSP | 68.94 | 86.41 | 78.11 | 77.35 | 88.19 | 42.42 | 85.06 | 56.44 | 84.87 | 56.11 |
| | OpenMax | 68.60 | 89.20 | 78.68 | 88.36 | 94.93 | 19.56 | 85.52 | 73.17 | 87.36 | 73.74 |
| | ReAct | 63.10 | 90.46 | 75.43 | 78.50 | 86.11 | 48.22 | 86.66 | 55.87 | 84.29 | 57.68 |
| | RMDS | **72.87** | 84.53 | **87.31** | **46.22** | **96.10** | 19.46 | 89.38 | 37.23 | 92.32 | 29.57 |
| | SHE | 68.04 | 85.74 | 84.18 | 56.01 | 93.57 | 22.17 | **92.65** | 25.65 | 91.04 | 33.59 |
| | ViM | 69.42 | 90.06 | 84.64 | 57.45 | 95.72 | **17.59** | 90.61 | 40.41 | 92.18 | 29.59 |
| | TempScale | 68.55 | 87.36 | 77.80 | 81.90 | 88.54 | 43.08 | 85.39 | 58.22 | 85.04 | 60.00 |
| | **Ours** | 70.03 | 83.99 | 84.68 | 49.27 | 93.40 | 22.55 | 91.00 | 32.43 | 91.34 | **29.49** |
| Swin-T | ASH | 46.28 | 95.22 | 47.00 | 93.98 | 46.49 | 94.19 | 41.32 | 96.11 | 45.22 | 93.99 |
| | GEN | **72.78** | **79.35** | **85.16** | **48.10** | 94.23 | 20.47 | 88.16 | 45.04 | 90.60 | 32.77 |
| | GradNorm | 50.43 | 93.37 | 45.01 | 94.06 | 38.28 | 95.18 | 34.75 | 97.80 | 34.00 | 96.83 |
| | KNN | 64.21 | 85.08 | 79.16 | 58.14 | 88.91 | 31.15 | 90.56 | 35.79 | 88.62 | 36.30 |
| | MDS | 68.69 | 83.72 | 81.78 | 53.55 | 93.60 | 21.89 | 89.82 | 37.47 | 90.98 | 30.89 |
| | MSP | 71.75 | 81.02 | 81.69 | 60.36 | 89.84 | 37.50 | 83.27 | 61.49 | 85.81 | 48.97 |
| | OpenMax | 71.52 | 85.54 | 81.69 | 72.85 | 95.06 | 19.56 | 82.81 | 76.41 | 87.75 | 60.88 |
| | ReAct | 69.36 | 85.01 | 82.12 | 60.08 | 90.08 | 31.34 | 87.04 | 53.98 | 87.85 | 42.54 |
| | RMDS | 71.81 | 82.66 | 84.91 | 49.89 | **95.57** | 18.55 | 89.26 | 39.09 | 92.10 | 29.34 |
| | SHE | 70.75 | 85.03 | 82.74 | 67.53 | 92.78 | 33.25 | 88.86 | 51.32 | 88.04 | 52.88 |
| | TempScale | 71.83 | 82.12 | 82.09 | 63.02 | 90.36 | 36.84 | 83.68 | 62.89 | 86.20 | 50.30 |
| | **Ours** | 69.17 | 82.75 | 82.90 | 54.00 | 93.17 | 23.27 | 90.41 | 35.17 | 91.31 | 30.84 |

Table 1: OOD detection performance on ImageNet-1K benchmarks, evaluated on a ViT-B/16 and a Swin Transformer. For each metric within each backbone, the best result is in **bold** and the second and third results are underlined. Detailed experimental results for all benchmarks are provided in Appendix D.

These results support our core hypothesis: measuring structural deviation from an explicitly learned model of ID features provides a more fundamental and reliable OOD signal than conventional confidence scores or feature space density measures. By decomposing ViT's dense [CLS] representations into meaningful sparse space using a SAE, our method can reference the stable, class-conditioned activation patterns of ID data to detect OOD inputs with subtly different structures. Overall, this approach offers a new paradigm for OOD detection that combines high accuracy, strong FPR95 robustness, and interpretability, while remaining computationally efficient and free of backbone retraining.

## 4.3 RESULTS ON OTHER BACKBONES

To substantiate the generalizability of our proposed method, we extended our evaluation to two additional Transformer-based backbones: Swin Transformer and DINOv2. These experiments validate that our approach is not tailored to a specific architecture but serves as a robust, model-agnostic method for OOD detection.

As shown in Table 1, our method maintains strong and competitive performance when applied to the Swin Transformer demonstrating consistent efficacy across all benchmarks. This result confirms that the core principle of our method is fundamentally compatible with different attention mechanisms and architectural designs within the Transformer family.

| Dataset | AUROC | FPR95 | AUPR IN | AUPR OUT |
|---|---|---|---|---|
| SSB hard | 74.57 | 81.38 | 69.84 | 77.47 |
| NINCO | 88.76 | 48.68 | 98.19 | 56.09 |
| iNaturalist | 98.22 | 6.98 | 99.61 | 91.61 |
| Textures | 91.71 | 38.16 | 98.87 | 63.83 |
| OpenImage-O | 95.62 | 19.37 | 98.41 | 88.49 |

Table 2: OOD detection performance on ImageNet-1K benchmarks, evaluated on DINOv2-ViT-B. Note that DINOv2 utilizes average-pooled patch embeddings, diverging from the [CLS] token in other backbones.

More strikingly, when paired with the DINOv2 backbone, our method's performance is not merely upheld but significantly enhanced, as detailed in Table 2. The improvements are particularly dramatic on challenging Far-OOD datasets. This substantial performance gain suggests that our approach is highly synergistic with

the rich, high-quality representations learned by advanced self-supervised models. The more semantically meaningful features from DINOv2 likely allow our Sparse Autoencoder to model the in-distribution manifold with even greater precision, making out-of-distribution deviations more apparent.

In summary, these results confirm that our SAE approach is a versatile and powerful OOD detection method that generalizes effectively across diverse Transformer architectures. Furthermore, its performance scales with the quality of the backbone's feature extractor, highlighting its potential for future applications with even more advanced models.

## 5 CONCLUSION

SAE has recently proven effective for feature interpretability in large language models, revealing sparse, human-readable factors within otherwise dense representations. In this work, we show that the same idea transfers successfully to vision: training a SAE on [CLS] tokens "unfolds" representations into a sparse latent space where class-specific structure becomes explicit and stable across exemplars. Within this space, a compact head of features consistently carries most activation while the tail remains largely silent, yielding a clear, class-conditioned view of the underlying representation structure.

Leveraging the class-conditioned structure exposed by the SAE, we define CAPs and suggest a new OOD detection method using Spearman rank correlation, which tests whether the canonical rank hierarchy based on trained ID data is preserved. Without modifying or retraining the ViT backbone, these scores achieve competitive AUROC scores with recent benchmark methods and set a new SOTA on FPR95 on the OpenImage-O dataset, a standard challenging OOD benchmark dataset, while showing uniform robustness across datasets. Unlike several existing approaches that fluctuate dramatically across datasets, our method remains consistently robust. We attribute this stability to testing conformity to the intrinsic feature structure rather than relying on post-hoc heuristics or complex boundary learning.

To sum up, we adapt SAEs for vision interpretability for the first time and define a class-wise activation signature — a fixed ranking of latent dimensions that captures the typical structure of each class and serves as a reference for OOD analysis. Drawing from these findings, our method enables simple yet robust OOD detection, reaching state-of-the-art robustness and consistency.

Moreover, this work shows that sparsity-driven analysis is not limited to language: applying SAEs to ViT yields a transparent, class-conditioned representation that is not merely interpretable but also practical for reliable OOD detection. Uncovering and leveraging the intrinsic structure of ViT features with SAEs offers a simple yet powerful path toward more trustworthy vision systems.

**Limitations**. In this study, we employ a standard vanilla SAE whose training objective is optimized for reconstruction performance rather than explicitly encouraging class-discriminative latent features. Consequently, there is room to further improve OOD detection by optimizing the SAE to better disentangle class-specific factors. Additionally, CAPs are estimated from ID data and can reflect class imbalance or distributional drift. Finally, our analysis focuses on [CLS] tokens in classification settings; extending the approach to patch tokens and to dense prediction tasks (e.g., detection and segmentation) remains for future work.

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

# A    OOD ACTIVATION MEAN AT CORE INDICES

Figure 6 extends the analysis from Figure 3 to additional OOD datasets. For each dataset, we select representative ImageNet classes and plot the activation distributions of ID (blue) and OOD (red) samples over the core indices of those classes. The left column shows OOD samples compared to ID on the core indices of their predicted class, while the right column shows the same OOD samples on the core indices of unrelated classes. Across all datasets, a consistent pattern emerges: OOD inputs fire the core indices of the class they are assigned to more strongly than unrelated ones, but their activation strength remains systematically below that of true ID samples.

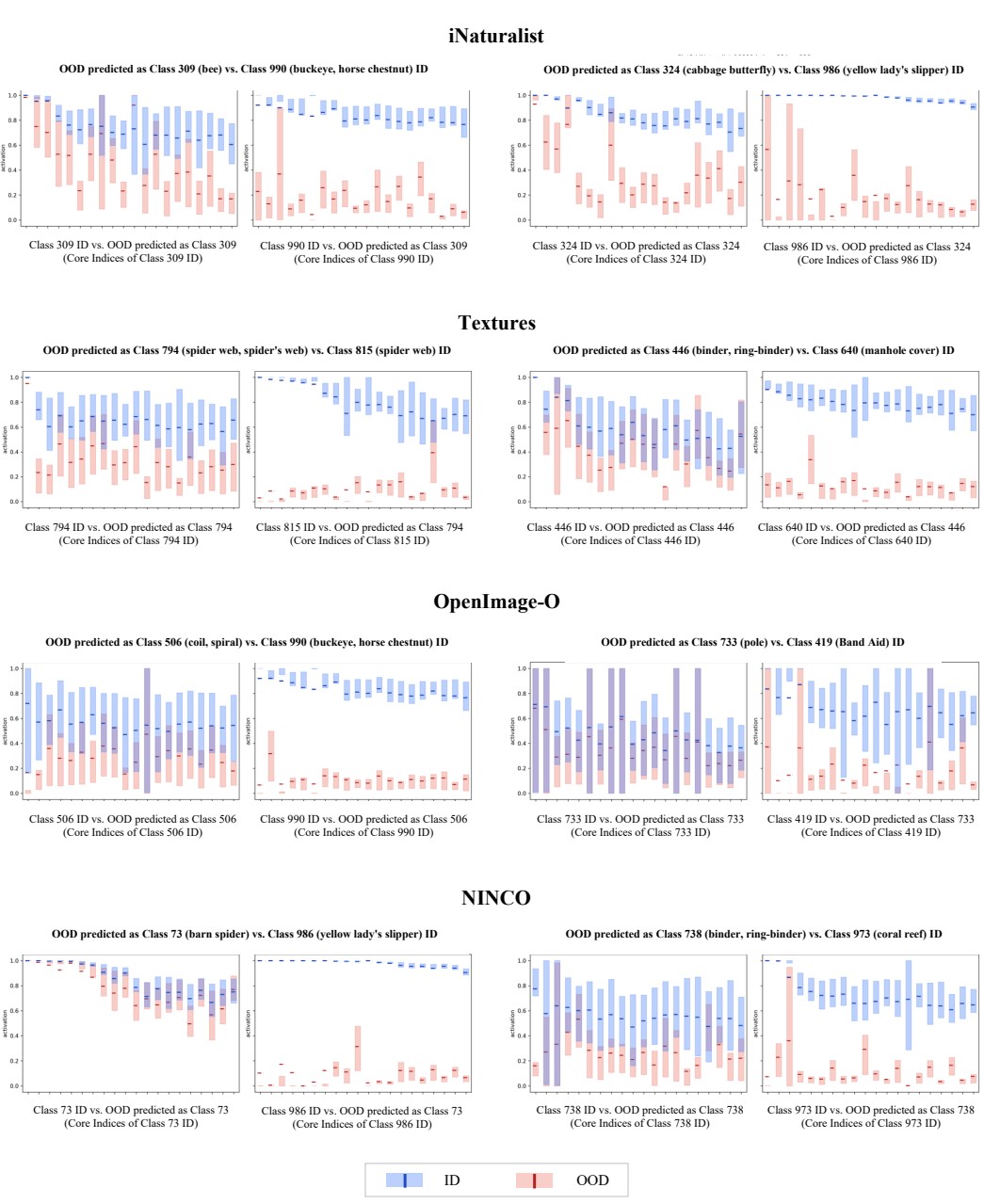

Figure 6: ID vs OOD activation comparison at core indices across multiple OOD datasets. From top to bottom: iNaturalist, Textures, OpenImage-O, and NINCO. Each pair of plots compares ID samples (blue) and OOD samples (red) on the core indices of the class the OOD samples were predicted as (left) and on unrelated classes (right)

## B ID VS OOD VS OOD-OTHERS CORE INDEX COMPARISON

These plots extend Fig. 4 to other OOD datasets, confirming the same trend: ID samples show the highest activation on their own class's core indices, OOD samples predicted as a given class activate its core indices but with reduced strength, and activations on unrelated classes remain lowest.

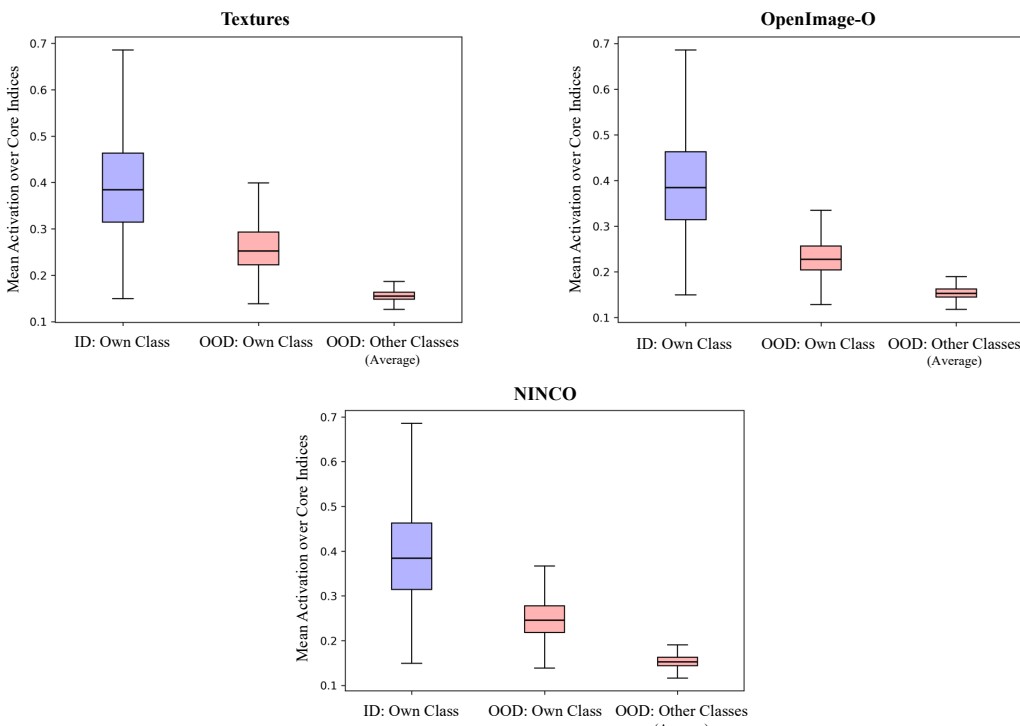

Figure 7: ID vs OOD activation comparison at class-specific core indices across additional OOD datasets. Each group of three boxplots follows the same format as Fig. 4: left — ID samples on their own class's core indices; middle — OOD samples on the core indices of the class they were predicted as; right — OOD samples on the core indices of unrelated classes. Shown here are results for Textures, OpenImage-O, and NINCO, complementing the iNaturalist example in the main paper.

# C CLASS-WISE ACTIVATION DISTRIBUTION ANALYSIS FOR ID AND OOD SAMPLES

This section includes extension of the class-wise activation–profile analysis of Fig. 5 to additional OOD datasets. For each dataset, we select representative classes and plot activation profiles along the CAP-ordered dimensions of the predicted ImageNet class. The x-axis follows the per-class CAP (highest-to-lowest ID mean activation); the y-axis shows the corresponding activations for ID and OOD samples.

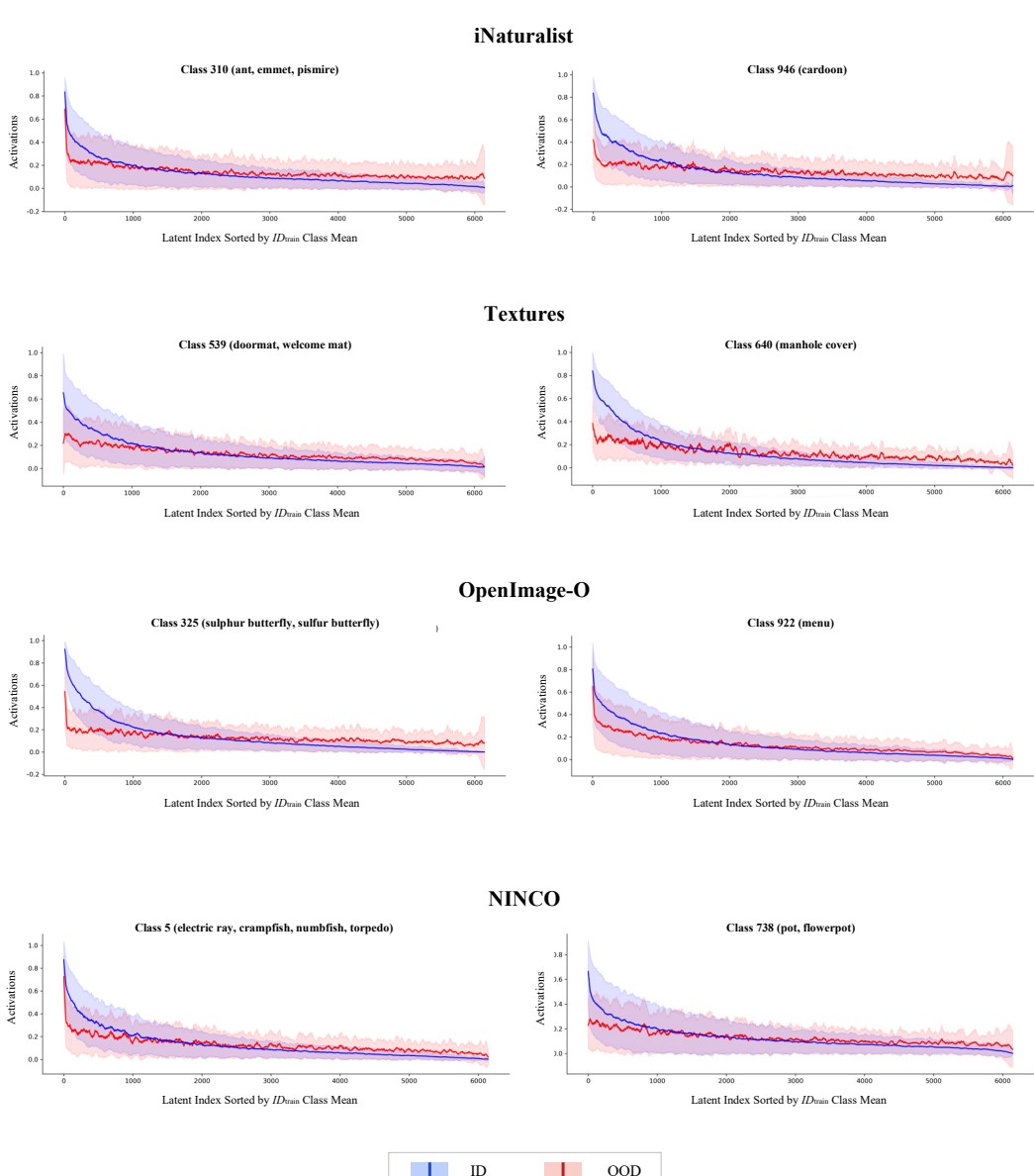

Figure 8: CAP-ordered activation profiles across additional OOD datasets across Textures, OpenImage-O, and NINCO

# D  FULL EXPERIMENTAL RESULTS

## D.1  FULL RESULTS ON VIT

| Method | Near-OOD | | | | Far-OOD | | | | | |
|---|---|---|---|---|---|---|---|---|---|---|
| | SSB-Hard | | NINCO | | iNaturalist | | Textures | | OpenImage-O | |
| | AUROC | FPR95 | AUROC | FPR95 | AUROC | FPR95 | AUROC | FPR95 | AUROC | FPR95 |
| ASH | 53.89 | 93.50 | 52.52 | 95.40 | 50.63 | 97.02 | 48.53 | 98.49 | 55.52 | 94.80 |
| DICE | 59.05 | 89.77 | 71.67 | 81.09 | 82.51 | 47.92 | 82.21 | 54.79 | 82.23 | 52.57 |
| EBO | 58.80 | 92.25 | 66.02 | 94.16 | 79.30 | 83.58 | 81.17 | 83.65 | 76.48 | 88.79 |
| GEN | 70.09 | 82.24 | 82.51 | 59.31 | 93.54 | 22.94 | 90.23 | 38.31 | 90.27 | 35.43 |
| GradNorm | 42.96 | 93.62 | 64.40 | 95.81 | 42.42 | 91.16 | 44.99 | 92.25 | 37.82 | 94.52 |
| KNN | 65.97 | 86.22 | 82.25 | 54.73 | 91.46 | 27.74 | 91.12 | 33.23 | 89.86 | 34.82 |
| MDS | 71.57 | 83.47 | 86.52 | 48.76 | 96.01 | 20.66 | 89.41 | 38.90 | 92.38 | 30.35 |
| MLS | 64.20 | 91.52 | 72.40 | 92.98 | 85.29 | 72.98 | 83.74 | 78.93 | 81.60 | 85.78 |
| MSP | 68.94 | 86.41 | 78.11 | 77.35 | 88.19 | 42.42 | 85.06 | 56.44 | 84.87 | 56.11 |
| OpenMax | 68.60 | 89.20 | 78.68 | 88.36 | 94.93 | 19.56 | 85.52 | 73.17 | 87.36 | 73.74 |
| ReAct | 63.10 | 90.46 | 75.43 | 78.50 | 86.11 | 48.22 | 86.66 | 55.87 | 84.29 | 57.68 |
| RMDS | 72.87 | 84.53 | 87.31 | 46.22 | 96.10 | 19.46 | 89.38 | 37.23 | 92.32 | 29.57 |
| SHE | 68.04 | 85.74 | 84.18 | 56.01 | 93.57 | 22.17 | 92.65 | 25.65 | 91.04 | 33.59 |
| TempScale | 68.55 | 87.36 | 77.80 | 81.90 | 88.54 | 43.08 | 85.39 | 58.22 | 85.04 | 60.00 |
| ViM | 69.42 | 90.06 | 84.64 | 57.45 | 95.72 | 17.59 | 90.61 | 40.41 | 92.18 | 29.59 |
| Ours | 70.03 ±0.03 | 83.99 ±0.04 | 84.68 ±0.04 | 49.27 ±0.26 | 93.40 ±0.14 | 22.55 ±0.08 | 91.00 ±0.01 | 32.43 ±0.05 | 91.34 ±0.01 | 29.49 ±0.04 |

Table 3: OOD detection performance on ImageNet-1K benchmarks, evaluated on a ViT-B/16. For each metric, the best result is in **bold** and the second and third results are underlined. Results for our method are the average of 5 runs.

## D.2  FULL RESULTS ON SWIN TRANSFORMER

| Method | Near-OOD | | | | Far-OOD | | | | | |
|---|---|---|---|---|---|---|---|---|---|---|
| | SSB-Hard | | NINCO | | iNaturalist | | Textures | | OpenImage-O | |
| | AUROC | FPR95 | AUROC | FPR95 | AUROC | FPR95 | AUROC | FPR95 | AUROC | FPR95 |
| ASH | 46.28 | 95.22 | 47.00 | 93.98 | 46.49 | 94.19 | 41.32 | 96.17 | 45.22 | 93.99 |
| DICE | 49.97 | 95.76 | 50.00 | 97.28 | 47.09 | 98.11 | 77.61 | 87.37 | 58.67 | 96.36 |
| EBO | 68.28 | 87.51 | 78.42 | 79.23 | 85.17 | 61.18 | 79.00 | 84.29 | 80.24 | 80.20 |
| GEN | 72.78 | 79.35 | 85.16 | 48.10 | 94.23 | 20.47 | 88.16 | 45.04 | 90.60 | 32.77 |
| GradNorm | 50.43 | 93.37 | 45.01 | 94.06 | 38.28 | 95.18 | 34.75 | 97.80 | 34.00 | 96.83 |
| KNN | 64.21 | 85.08 | 79.16 | 58.14 | 88.91 | 31.15 | 90.56 | 35.79 | 88.62 | 36.30 |
| MDS | 68.69 | 83.72 | 81.78 | 53.55 | 93.60 | 21.89 | 89.82 | 37.47 | 90.98 | 30.89 |
| MLS | 70.47 | 86.60 | 80.95 | 75.78 | 89.01 | 49.65 | 81.70 | 79.94 | 83.93 | 73.96 |
| MSP | 71.75 | 81.02 | 81.69 | 60.36 | 89.84 | 37.50 | 83.27 | 61.49 | 85.81 | 48.97 |
| OpenMax | 71.52 | 85.54 | 81.69 | 72.85 | 95.06 | 19.56 | 82.81 | 76.41 | 87.75 | 60.88 |
| ReAct | 69.36 | 85.01 | 82.12 | 60.08 | 90.08 | 31.34 | 87.04 | 53.98 | 87.85 | 42.54 |
| RMDS | 71.81 | 82.66 | 84.91 | 49.89 | 95.57 | 18.55 | 89.26 | 39.09 | 92.10 | 29.34 |
| SHE | 70.75 | 85.03 | 82.74 | 67.53 | 92.78 | 33.25 | 88.86 | 51.32 | 88.04 | 52.88 |
| TempScale | 71.83 | 82.12 | 82.09 | 63.02 | 90.36 | 36.84 | 83.68 | 62.89 | 86.20 | 50.30 |
| ViM | 68.94 | 88.55 | 81.85 | 60.80 | 94.62 | 17.98 | 92.69 | 29.46 | 92.29 | 26.62 |
| Ours | 69.17 ±0.02 | 82.75 ±0.03 | 82.90 ±0.06 | 54.00 ±0.20 | 93.17 ±0.05 | 23.27 ±0.15 | 90.41 ±0.03 | 35.17 ±0.18 | 91.31 ±0.03 | 30.84 ±0.09 |

Table 4: OOD detection performance on ImageNet-1K benchmarks, evaluated on a Swin Transformer. For each metric, the best result is in **bold** and the second and third results are underlined.

# E  DATASET DETAILS

## E.1  IN-DISTRIBUTION (ID) DATASET

**ImageNet-1K.** In this study, we use ImageNet-1K (Deng et al. (2009)) as the In-Distribution (ID) dataset. ImageNet-1K is widely recognized as the standard benchmark for pre-training and evaluating object recognition models in computer vision. The dataset consists of approximately 1.28 million training images and 50,000 validation images across 1,000 object classes. Each class represents a specific object, such as 'balloon' or 'strawberry,' or a particular animal species like 'African elephant.' The vast scale and high intra- and inter-class diversity of ImageNet-1K enable models to learn rich, generalizable visual features, providing a solid foundation for OOD detection research.

## E.2  OUT-OF-DISTRIBUTION (OOD) DATASETS

Out-of-Distribution (OOD) datasets contain data from novel distributions not seen during training and are used to evaluate a model's ability to reject unfamiliar inputs. In our work, we utilize a variety of OOD datasets with varying semantic distances from the ID dataset (ImageNet-1K) to comprehensively assess the robustness of our method.

**NINCO.** NINCO (Bitterwolf et al. (2023)) is a challenging benchmark specifically designed for OOD detection. It comprises near-OOD samples that are visually similar but semantically unrelated to the ImageNet classes, making it highly effective for evaluating a model's ability to distinguish subtle differences. For instance, while ImageNet may contain a 'baseball' class, NINCO could provide images of 'cricket balls' as near-OOD examples.

**SSB-Hard (Semantic Shift Benchmark-Hard).** SSB-Hard (Vaze et al. (2022)) is an OOD dataset composed of new images corresponding to the 1,000 classes in ImageNet-1K. However, these images have undergone a semantic shift, presenting different styles, poses, or environments compared to the original ImageNet images. It serves as a near-OOD dataset to test how fundamentally a model understands the class concepts it has learned.

**iNaturalist.** iNaturalist (Van Horn et al. (2018)) is a fine-grained classification dataset containing images of diverse plant and animal species. The iNaturalist test set used for OOD benchmarks is carefully constructed by selecting novel species from the massive original dataset which contains tens of millions of images that do not overlap with any of the 1,000 classes in ImageNet-1K. It includes species that are visually similar to those in ImageNet but are much more specific, testing the model's ability to detect fine-grained feature differences.

**OpenImage-O.** OpenImage-O (Wang et al. (2022)) is an OOD benchmark extracted from the large-scale Open Images dataset. It includes objects that do not overlap with the ImageNet-1K classes and features a mix of various objects and scenes, providing naturally distributed far-OOD data that resembles real-world scenarios.

**Textures (DTD).** The Describable Textures Dataset (Cimpoi et al. (2014)) consists of images representing 47 different texture patterns, such as 'striped', 'dotted',, and 'interlaced'. As it focuses on texture and patterns rather than object shapes, it is useful for evaluating how a model perceives general visual patterns beyond specific object classes.

# F  OOD SCORE CALCULATION DETAILS

The feature vector $h$ is extracted from the pre-trained backbone model, where $h = f_{\text{backbone}}(x)$. For ViT, $h$ corresponds to the [CLS] token representation, while for ResNet-50, it is the output of the final average pooling layer. This feature vector is then passed through our trained SAE encoder, $f_{\text{enc}}$, to produce the latent activation vector:

$$a = \sigma\left(f_{\text{enc}}\left(f_{\text{backbone}}(x)\right)\right) \tag{2}$$

The final OOD score is the Spearman's rank correlation coefficient ($\rho$) between the rank vector of the sample's activation, $R_a$, and the pre-computed Class Activation Profile (CAP) for class $c$, denoted as $R_c$. The score is computed using the standard formula:

$$\text{Score}(x) = \rho(R_a, R_c) = 1 - \frac{6 \sum d_i^2}{n(n^2 - 1)} \tag{3}$$

Here, $d_i$ represents the difference between the ranks of the two vectors ($R_a$ and $R_c$) for each neuron $i$, and $n$ is the total number of latent neurons. This score ranges from $-1$ to $1$.

## THE USE OF LARGE LANGUAGE MODELS (LLMS)

We used a large language model (LLM), specifically GPT-5, primarily as an editing and translation tool to polish the English phrasing and grammar of this manuscript. The LLM's role was strictly limited to refining the language of the text that was already drafted by the authors. It was used to assist with a variety of tasks, including:

- Translation assistance: The model helped to translate and refine sections of the manuscript that were originally drafted in Korean.

- Grammar and syntax correction: The LLM was used to correct grammatical errors and improve the overall flow and readability of the English text.

- Clarity and phrasing enhancement: The model helped to rephrase sentences to ensure they were clear, concise, and academically appropriate.

- Verify that the English phrasing was natural and idiomatic, helping us to avoid awkward or stilted expression.

The LLM played no significant role in the research ideation, experimental design, data analysis, or the generation of the core content and conclusions of this study. All research findings, arguments, and intellectual contributions presented in this paper are the original work of the authors.

