# OpenReview forum: "Sparsity Reveals Strangers: A Sparse Autoencoder Approach to OOD Detection"
_ICLR.cc/2026/Conference — ICLR 2026 Conference Withdrawn Submission_

### Official Review · Reviewer_nQyH · 2025-10-29

**Soundness:** 3
**Presentation:** 3
**Contribution:** 3
**Rating:** 6
**Confidence:** 4

**Summary:**

This paper applies a Sparse Autoencoder, commonly used in the context of LLM interpretability, to a Vision Transformer (ViT) for out-of-distribution (OOD) detection. The authors first conduct a thorough analysis of the Sparse Autoencoder’s behavior on both in-distribution (ID) and OOD data, and then build their proposed method based on these insights. Experimental results show that the approach achieves comparable or even superior performance to existing state-of-the-art post-hoc detection methods, demonstrating the promise of applying Sparse Autoencoders to OOD detection.

**Strengths:**

- S1. This paper applies the concept of Sparse Autoencoders, originally used in the context of model interpretability, to the OOD detection problem. This represents a distinct and novel approach compared to existing detection methods.

-  S2. Before introducing the proposed approach, Section 3 (Pattern Analysis) presents a detailed examination showing that the Sparse Autoencoder can successfully disentangle class information. This analysis supports the validity of the proposed method, which is an excellent aspect of the paper.

- S3. The experimental results demonstrate that the proposed method achieves comparable or even superior performance to other state-of-the-art OOD detection techniques. This provides strong evidence that the Sparse Autoencoder-based approach is effective for OOD detection.

- S4. The paper also evaluates performance on near-OOD benchmarks, such as NINCO and OpenImage-O, as well as far-OOD datasets. Reporting results on both near and far OOD settings is highly commendable and adds robustness to the evaluation.

**Weaknesses:**

- W1. The score function is somewhat unclear. Although the mathematical formulation is provided in Appendix F, presenting this content in the main text would make it easier for readers to understand the proposed approach.

- W2. While using a Sparse Autoencoder is an interesting idea, it also introduces additional training cost. Most of the comparison methods in this paper are post-hoc approaches that can be applied without any training. Therefore, it would be helpful if the paper included a discussion of the training cost, such as how much time or computational resources are required for training a Sparse Autoencoder.

**Questions:**

I would like to know the discussion on training cost for a Sparse Autoencoder.

---

### Official Review · Reviewer_5yVa · 2025-10-29

**Soundness:** 3
**Presentation:** 3
**Contribution:** 2
**Rating:** 4
**Confidence:** 3

**Summary:**

This paper proposes a new out-of-distribution (OOD) detection method for pretrained Vision Transformers (ViTs).

The key idea builds on recent findings that sparse autoencoders (SAEs) produce latent spaces well aligned with the semantic structure of data. The authors apply an SAE to the [CLS] token embeddings of a pretrained ViT, mapping them into a sparse latent space. In this space, each in-distribution (ID) class exhibits a characteristic pattern where a small subset of dimensions are strongly activated. These class-specific activation patterns are summarized as Class Activation Profiles (CAPs). At test time, the activation pattern of a sample is compared with the CAP of the most confident class: samples whose activation structure closely matches the CAP are regarded as ID, while those with deviating activations are classified as OOD.

Experiments conducted on the OpenOOD v1.5 benchmark (ImageNet-1K as ID) show that the proposed method achieves competitive or superior results compared to exisitng OOD detection methods based on ViT backbones.

**Strengths:**

**S1.** Using SAEs for interpretability has already been explored, and this paper effectively transfers that idea to ViTs for OOD detection. While the overall approach is conceptually straightforward and may be somewhat viewed as an adaptation of existing ideas, the novelty lies in detecting OOD samples not through probabilistic scores or feature distances, but through structural deviations in activation patterns within a sparse latent space, which is a perspective that has rarely been explored in OOD detection literature.

**S2.** Sec. 3 provides compelling evidence that no ID classes share similar activation profiles with ImageNet-1K classes, and that ID and OOD samples exhibit clearly different activation distributions. This observation supports the key hypothesis in this paper and adds interpretability to the OOD decision mechanism.

**S3.** Experimental results on the OpenOOD v1.5 benchmark (ImageNet-1K as ID) show that the method achieves competitive or superior performance compared to the existing ViT-based OOD detection methods, while maintaining high interpretability.

**S4.** The paper is mostly well-written and clearly organized, making the methodology and insights easy to follow.

**Weaknesses:**

**W1. Novelty**

**W1-1.** The core idea builds on existing studies that use SAEs to interpret the internal representations (e.g., Cunningham et al. (2023)). This work adapts that concept to ViTs for OOD detection. While applying this idea to the ViT's [CLS] token and the OOD detection context is new, it largely follows the intuition established in prior interpretability research, and the conceptual leap is somewhat limited.

**W1-2.** The idea and empirical findings are interesting, but the algorithmic novelty is modest. No new learning objective or optimization strategy is introduced beyond applying SAE and computing activation similarity.


**W2. Training Data for ViT and SAE**

**W2-1.** SAE needs to be trained on ID data. In practice, however, access to ID samples cannot always be assumed. Since many strong post-hoc OOD detection methods operate entirely without retraining, this requirement could limit the applicability of the proposed approach in real-world settings.

**W2-2.** Although the ViT models remain frozen, the method still requires training an additional SAE from scratch. This introduces extra computational cost and dependence on training data, which somewhat weakens the claim of efficiency compared to purely post-hoc methods.

**W2-3.** According to the paper, ViT-B/16 and Swin follow OpenOOD's standard implementations (Zhang et al., NeurIPS 2023), which are pretrained on ImageNet-1K. Thus, the training data for SAE and the pre-trained data for ViT are aligned (ImageNet-1K). However, if the SAE were trained on data drawn from a different distribution, its learned activation patterns and resulting OOD detection accuracy could change.
A robustness analysis showing how performance varies when the SAE is trained on mismatched or limited data would strengthen the empirical validity of the method.


**W3. Evaluation Scope**

The evaluation is conducted solely under the ImageNet-1K ID setting. While ImageNet-1K is a widely used and challenging dataset, evaluating only this setting limits the demonstrated generality of the proposed method. OOD detection performance can vary substantially across ID datasets with different granularity and scale (e.g., CIFAR-10/100, ImageNet-200, iNaturalist). Since OpenOOD already supports multiple ID configurations, including experiments on at least one additional dataset would provide stronger evidence of robustness and general applicability.

**Questions:**

**For W1**. If the above understanding is incorrect, please clarify.

**For W2**

- Could the authors discuss how the method behaves when access to the ID dataset is limited or partially unavailable?

- What is the computational cost for training SAEs? Is it significant? Would it be possible to pretrain the SAE on non-ID data?

- How might training SAE on different datasets affect CAP formation and OOD detection accuracy? How does performance change if the SAE is trained on data drawn from a distribution different from the ViT's pretraining data?

**For W3**. Could the authors report results on more diverse ID datasets (e.g., CIFAR-10/100, ImageNet-200) to demonstrate the method's generality across scales and domains?

---

### Official Review · Reviewer_h7AX · 2025-10-31

**Soundness:** 2
**Presentation:** 1
**Contribution:** 2
**Rating:** 2
**Confidence:** 4

**Summary:**

The work proposes a sparse autoencoder-based framework using the vision transformer embeddings that will convert the dense representations to the sparse vectors. By inspecting the per-class activation profiles (CAPs) of the latent embeddings, the work introduces an OOD score to detect the ID class vs OOD during the test time. Empirical evaluations are presented to showcase the effectiveness of the method.

**Strengths:**

Strengths:

•	The observations that the head/tail differences between ID and OOD in the latent activation profile looks interesting. The plots clearly shows the gap in mean activation value.

**Weaknesses:**

Please see Questions section.

**Questions:**

Weaknesses:

1.	The main drawback is the limitation in novelty of the work. The idea of exploiting sparsity in latent dimensions are not entirely novel. For example, see the work [1]. The use of sparse autoencoders is interesting, but it is not well-justified why it should be preferred over the existing methods.

2.	In the introduction, there is a sentence reasoning out the limitations of the existing work “While this token provides a rich, holistic representation of the input, its dense and entangled nature makes it a “black box”…”. This is very vague argument than specifically pointing out the existing limitations.

3.	The OOD detection score depends on the predicted class. But this implies that the OOD detection score is highly dependent on the classification performance, which could have been easily avoided using the pre-trained feature representations from ViT. This is typically done is many CLIP-based prompt tuning OOD methods that relies on the feature representations. Here, that information is completely unexploited.

4.	Choice Spearman correlation coefficient for being robust to outliers does not seem correct and well-justified. Could you provide some citation or any recent results that support this claim, as I am not aware of this?

5.	The presentation of the idea also lacks technical depth. SAE is not introduced in detail which is the main component; also the latent representations and its dimensions are all not well-defined.

6.	As the ViT is pre-trained, does your approach require training on entire data on the ImageNet-1K dataset? Or does it support few-shot training? If that is the case, how many samples per class is required?

7.	The main Table (Table 1) also does not justify the effectiveness of the approach over the baselines. Also, not many recent baselines are compared here.

Overall, the paper lacks technical depth, novelty and well-substantiating empirical evidence.

[1] Ghosal, Soumya Suvra, Yiyou Sun, and Yixuan Li. "How to overcome curse-of-dimensionality for out-of-distribution detection?." Proceedings of the AAAI Conference on Artificial Intelligence. Vol. 38. No. 18. 2024.

---

### Official Review · Reviewer_18HD · 2025-11-01

**Soundness:** 2
**Presentation:** 2
**Contribution:** 2
**Rating:** 4
**Confidence:** 4

**Summary:**

This paper introduces a sparse autoencoder (SAE)–based framework for interpreting and detecting out-of-distribution (OOD) samples in Vision Transformers (ViTs).
The key idea is to train an SAE on the [CLS] representations of in-distribution (ID) data, unfolding the dense features into a sparse latent space where class-specific activation patterns emerge.
Each class exhibits a Class Activation Profile (CAP) — a stable rank ordering of latent activations.
At test time, the Spearman rank correlation between a sample’s latent activation and the CAP of its predicted class serves as an OOD score.
Empirically, this simple metric achieves competitive or state-of-the-art OOD performance on multiple benchmarks (iNaturalist, Textures, OpenImage-O, NINCO) across different backbones (ViT-B/16, Swin-T, DINOv2).
The method is architecture-agnostic, interpretable, and computationally light.

**Strengths:**

- Conceptually interesting idea:
Connecting sparse autoencoder interpretability with OOD detection offers a novel and intuitive perspective.

- Empirical validation:
Results are competitive and consistent across ViT, Swin, and DINOv2 backbones.

- Readable and well-structured presentation:
Figures and analyses clearly illustrate the emergence of class-specific sparse activation patterns.

**Weaknesses:**

- Limited theoretical depth:
Lacks a formal justification for why CAP rank consistency should correlate with OOD likelihood.

- Modest performance margin:
Results are close to, but not clearly above, strong baselines like RMDS and GEN.

- Scope restriction:
Focused only on ViT-based features; unclear if applicable to CNNs or multimodal encoders.

- Ablation clarity:
1) Some hyperparameter choices (e.g., sparsity ratio, CAP truncation) appear empirical without deeper analysis.
2) Comparing the proposed SAE with alternative autoencoder variants (e.g., VAE, Denoising AE) -- whether the gains truly stem from sparsity rather than generic reconstruction.
3) Contrasting the rank-correlation score with distribution-based metrics such as KL divergence or Mahalanobis distance.

- Other interpretability techniques:
(E.g., concept bottleneck models, sparse coding, CAM, TCAV), but does not discuss how its SAE-based structure compares to these in interpretability quality or computational cost.

- Unverified prototype averaging strategy:
The method relies on a single class-mean CAP as the prototype for computing rank correlation, but it is unclear whether this is optimal.
The paper does not examine alternative strategies such as using multiple representative CAPs or clustering within each class, which might better capture intra-class variation and improve robustness.

- Lack of ID classification accuracy:
The paper does not report the in-distribution (ID) classification accuracy of the proposed SAE + CAP framework.
Although the method is primarily designed for OOD detection, evaluating its ID classification performance (e.g., via rank-correlation-based prototype matching) would help verify whether the learned sparse representations preserve class-discriminative information.
The absence of such an analysis slightly limits the understanding of how much semantic structure is retained in the latent space.

- Limited analysis of SAE representations:
The paper does not provide concrete analyses of what individual SAE latent dimensions represent.
There is no deeper examination (e.g., visualizing which features or semantics each index captures), making the claimed interpretability somewhat abstract.

**Questions:**

- Could this method extend to localized OOD detection (e.g., spatial anomalies) or multimodal embeddings (e.g., CLIP)?

- How does the SAE training scale to other datasets or different pretraining distributions except only ImageNet?

- Would simpler baselines (e.g., PCA + cosine distance) achieve similar trends?

---

### Note · Authors · 2025-11-14

I have read and agree with the venue's withdrawal policy on behalf of myself and my co-authors.